# The Application of Bilayer Heterogeneous MOFs in pH and Heat-Triggered Systems for Controllable Fragrance Release

**DOI:** 10.3390/ma17061310

**Published:** 2024-03-12

**Authors:** Tianci Huang, Xinjiao Cui, Xiaoyu Zhou, Xiaolong He, Min Guo, Junsheng Li

**Affiliations:** 1School of Chemistry, Chemical Engineering and Life Sciences, Wuhan University of Technology, Wuhan 430070, China; 17807305139@163.com (T.H.); 304805@whut.edu.cn (X.C.); 318517@whut.edu.cn (X.Z.); 2Zernike Institute for Advanced Materials, University of Groningen, Nijenborgh 4, 9747 AG Groningen, The Netherlands; xiaolong.he@rug.nl; 3School of Mechanical and Electronic Engineering, Wuhan University of Technology, Wuhan 430070, China

**Keywords:** MOFs, bilayers, controllable release, fragrance, adsorption

## Abstract

To facilitate the integration of a fragrance encapsulation system into different products to achieve effective releases, a dual-responsive release system with pH and thermal trigger control is designed in this work. A series of ZIF-8 (M) and bilayer ZIF-8-on-ZIF-8 (MM) materials are synthesized by a solvent method at room temperature. The fragrance is encapsulated into the ZIFs by dynamic adsorption or in situ encapsulation combined dynamic adsorption. The fragrance loading contributed by dynamic adsorption was as high as 80%. The fragrance loaded in the double-layer MM host was almost twice that of the monolayer host M due to the stronger electrostatic interaction between MM and vanillin. In the pH and thermal trigger response release experiments, the second MOF layer in the MM host, as a controlled entity, greatly improved the load and kinetic equilibrium time of vanillin, and realized the controlled release of guest molecules. The developed dual-responsive release system in this work exhibits great potential in daily chemical products.

## 1. Introduction

In the formulation of any fragrance product or food, flavorings can be some of the most valuable ingredients. However, because of their inherent instability, preserving them is often a major concern for manufacturers [1]. To minimize degradation or loss of aroma during processing and storage, it is beneficial and necessary to encapsulate volatile compounds in food or other fragrance products. Encapsulation is a common technique used to protect and preserve biologically active, volatile, and easily degradable compounds, shielding them from degradation and helping to mask undesirable odors [2]. Approximately 60 years ago, encapsulation technology was first developed, with coatings that can control the release of compounds at specific rates under specific conditions [3,4].

So far, most commercially available fragrance capsules are prepared using physical methods, such as spray drying and freeze drying. However, it has been proven that capsules prepared by these methods have poor stability, which limits their broader applications. Additionally, the coating materials used in physical encapsulation processes are mostly organic materials such as polymers, cyclodextrins, liposomes, and carbohydrates. While these organic materials possess characteristics such as biodegradability, low toxicity, and chemical modifiability [5], they often have limitations in controlling the release of encapsulated molecules. Physical–chemical adsorption methods for preparing fragrance encapsulation systems involve simply embedding or mixing the guest molecules within the pores of porous materials. These systems, known as physical–chemical adsorption-based encapsulation, offer advantages such as simplicity, low cost, high loading capacity, high stability, and controlled release, making them a promising approach to fragrance encapsulation (Appendix A). However, the relatively low biocompatibility of traditional porous materials is often a concern [6].

In comparison to traditional materials, metal–organic frameworks (MOFs, Appendix A for abbreviations) possess many unique properties [7,8]. Their large pore volume, high porosity, and high surface area not only facilitate the higher loading of fragrances or guest molecules but also offer advantages such as high biocompatibility, good biodegradability, ease of functionalization, and higher water solubility. Additionally, fragrance molecules can be chemically bound or physically encapsulated within carriers through various interactions. They can form bio-MOFs (metal–bioorganic frameworks) through methods such as surface attachment, covalent bonding, pore encapsulation, and in situ encapsulation. MOFs have been extensively applied in biomedical applications, such as Zr-MOF-loaded naproxen as a carrier for specific intestinal delivery [9], Mg-MOF to facilitate rapid drug metabolism [10], and Ca-Sr-MOF-loaded tetracycline for antimicrobial use [11]. The excellent encapsulation properties of MOF materials make them a unique encapsulation system.

ZIF-8, as a zeolitic imidazole framework material, is self-assembled from physiological system components, Zn^2+^, and 2-methylimidazole, and has excellent biocompatibility [12]. Meanwhile, ZIF-8 exhibits good thermal stability due to the strong interaction between the core metal ions and the nitrogen atoms in the coordination ligand [13]. ZIF-8 has a high specific surface area, high porosity, and controllable size, and it is easy to synthesis, making it highly capable of loading functional materials [14]. In addition, ZIF-8 has excellent acid responsiveness and can be stabilized in physiological aqueous environments, while the zeolite imidazolium ester skeleton collapses when the pH is weakly acidic (pH 5–6), which facilitates the control of the release of the functional materials by adjusting the pH [15]. On the basis of the above characteristics, ZIF-8 is widely used in the biomedical field to encapsulate various functional materials, such as 5-fluorouracil [16], gentiopicroside [17], riboflavin-5-phosphate [18], 6-mercaptopurine [19], and metformin [20]. On the basis of these advantages of ZIFs and aiming to further increase fragrance loading and achieve controlled release, a series of ZIF-8 materials with single-layer and double-layer core–shell structures were synthesized as carriers for encapsulating vanillin using a simple room temperature solvent method. The double-layer core–shell structure of ZIF-8-on-ZIF-8 was confirmed through structural characterization and morphology analysis. The optimal adsorption conditions were determined by varying the initial concentration of vanillin. This study found that the dynamic adsorption method contributed to 80% of the loading capacity and that the double-layer carrier had a higher vanillin loading capacity, almost twice that of the single-layer carrier. In pH and thermally triggered release experiments, it was found that the second layer of MOF in the double-layer carrier significantly improved the vanillin loading and the kinetic equilibrium time for controlling vanillin release, enabling the effective release of the guest molecule. The double-layer ZIF-8-on-ZIF-8 carrier shows great potential in pH and thermally triggered dual-responsive release systems.

## 2. Materials and Methods

### 2.1. Materials and Chemicals

Fe(NO_3_)_3_·9H_2_O, 2-methylimidazole (2-MIm), and vanillin (99.9%) were provided by Aladdin. ZnNO_3_·6H_2_O, methanol (CH_3_OH), hydrochloric acid (HCl), and sodium hydroxide (NaOH) were bought from Sinopharm Chemical Reagent Co., Ltd. (Shanghai, China).

### 2.2. Preparation of Samples 

(1)Preparation of ZIF-8/Fe-ZIF-8/ZIF-8-on-ZIF-8/Fe-ZIF-8-on-ZIF-8:

According to the synthesis method reported in Reference [21], ZIF-8 was synthesized. A total of 1.7 g of hexa-aqueous zinc nitrate and 3.6 g of dimethylimidazole were used. Dimethylimidazole and hexa-aqueous zinc nitrate were slowly added to a 160 mL methanol solution under stirring at room temperature and continuously stirred for 24 h. After centrifugation, washing, and drying, ZIF-8 was obtained. Fe-ZIF-8 was synthesized using the same method, with a molar ratio of Zn to Fe of 10:0.5. The preparation of ZIF-8-on-ZIF-8 involved using the synthesized ZIF-8 as a template. After centrifugation and washing without drying, it was dispersed in a 160 mL methanol solution. Dimethylimidazole and hexa-aqueous zinc nitrate were taken in corresponding masses according to the feeding ratio of primary growth to secondary growth of 1:0.5, 1:1, 1:2, and 1:4, and slowly added to the methanol solution under stirring at room temperature for 24 h. After centrifugation, washing, and drying, the core–shell structure of ZIF-8-on-ZIF-8 material was obtained. The preparation of Fe-ZIF-8-on-ZIF-8 used Fe-ZIF-8 as a template and followed the same method as ZIF-8-on-ZIF-8.

(2)Preparation of vanillin@ZIF-8 (VM)/vanillin@ZIF-8-on-ZIF-8 (VMM)/vanillin@ZIF-8-on-vanillin@ZIF-8 (VMVM):

Approximately 80 mg of vanillin, 1.7 g of hexa-aqueous zinc nitrate, and 3.6 g of dimethylimidazole were used. The vanillin was added to a 160 mL methanol solution and stirred at room temperature for 5 min until the it was completely dissolved. Dimethylimidazole and hexa-aqueous zinc nitrate were then slowly added to the solution under stirring at room temperature for 24 h. After centrifugation, vanillin@ZIF-8 (VM) was obtained. Using the ZIF-8 obtained in step (1) as a substrate, after centrifugation and washing without drying, it was dispersed in a 160 mL methanol solution. Following the same method as the preparation of VM, the feeding ratio of primary growth to secondary growth was 1:2, which was used to obtain vanillin@ZIF-8-on-ZIF-8 (VMM). Using the same method as for the VMM, vanillin@ZIF-8 (VM) was used as a substrate to obtain vanillin@ZIF-8-on-vanillin@ZIF-8 (VMVM).

### 2.3. Characterizations

XRD data were collected using an D8 Advance X-ray diffractometer with a Cu K radiation wavelength of 1.5406 Å (40 kV, 50 mA), and the scanning range was 5–80° (10 min^−1^). A Zeiss Ultra Plus scanning electron microscope (SEM) was used to perform the measurements. The accelerating voltage was 20.0 kv. The internal structure of the double-layer Fe-ZIF-8-on-ZIF-8 was observed by a transmission electron microscope (TEM, JSM-2100F) from Nippon Electronics Co., Ltd. (Tokyo, Japan). The specific surface area analyzer ASAP2460 from the Micromeritics company was selected to test the ZIF-8 and the double-layer ZIF-8-on-ZIF-8 prepared with different feed ratios. The thermal stability of the single-layer ZIF-8 and double-layer ZIF-8-on-ZIF-8 before and after vanillin adsorption was analyzed by an STA449F3 integrated thermal analyzer. The American AVATAR FTIR-370 Fourier transform infrared spectrometer was selected to observe the interaction between ZIF-8 and ZIF-8-on-ZIF-8 and vanillin. The new Mastersizer 2000 laser particle size analyzer from the Malvern Company was selected to test ZIF-8 and double-layer ZIF-8-on-ZIF-8 prepared with different feed ratios, and the sizes and particle size distributions of different samples were observed. The zeta potential of vanillin, ZIF-8, and ZIF-8-on-ZIF-8 before and after vanillin adsorption was determined by a Zetasizer Nano ZS90 Zeta potentiometer from Malvern Co., Ltd. (WR14 1XZ., Fareham, UK).

### 2.4. Theoretical Calculations

To perform the structural optimization and density functional theory calculations, VASP software was used. The generalized gradient approximation (GGA) with the Perdew–Burke–Ernzerhof (PBE) functional was selected to calculate the exchange–correlation energy. The projector augmented-wave (PAW) pseudopotential method was used to describe the interaction between electrons and atomic nuclei, with the plane wave cut-off energy set to 500 eV. The convergence criteria for energy and atomic forces were set to 1 × 10^−5^ eV and 0.02 eV Å^−1^, respectively. The Monkhorst–Pack method was used to sample the first Brillouin zone, with k points set to 1 × 1 × 1. The van der Waals interactions were considered in the calculations. The binding energy was calculated using the formula:$$∆E=E_{total}+E_{ads}+E_{mat}$$

Here, $E_{total}$, $E_{mat}$, and $E_{ads}$ represent the energy of the system after adsorption, the energy of the optimized adsorbent material, and the energy of the adsorbate, respectively.

### 2.5. Encapsulation Performance Testing

A certain mass of vanillin was added to 20 mL of anhydrous ethanol to prepare vanillin solutions with concentrations of 0.3, 0.5, 0.7, 1.0, 1.5, 2.0, 2.5, and 3.0 mg mL^−1^. The solutions were stirred at room temperature for 10 min until the vanillin was completely dissolved. Then, 20 mg of ZIF-8 was added to each of the above solutions and stirred continuously at room temperature for 24 h. Each solution was divided into two equal portions. One portion was centrifuged and digested with hydrochloric acid, and the UV-visible absorption spectra at 280 nm were measured to determine the total adsorption capacity (TAC) of vanillin. The other portion was centrifuged, washed three times with ethanol, digested with hydrochloric acid, and the UV-visible absorption spectra were measured to determine the inner surface adsorption capacity (ISAC) of vanillin.

A total of 20 mg of ZIF-8 (denoted as M) and ZIF-8-on-ZIF-8 with different core–shell ratios (denoted as MM_0.5_, MM_1_, MM_2_, and MM_4_) were weighed and added to a vanillin solution (1 mg mL^−1^) and stirred continuously at room temperature for 24 h. The total adsorption capacity and inner surface adsorption capacity of each carrier were measured using UV-visible absorption spectroscopy.

The synthesized VM, VMM, and VMVM samples were digested with hydrochloric acid, and the UV-visible absorption spectra were measured. The measured total adsorption capacity is denoted as TAC-1, and the inner surface adsorption capacity is denoted as ISAC-1. The vanillin adsorption process was repeated using the same method to measure the total adsorption capacity (TAC-2) and inner surface adsorption capacity (ISAC-2) of vanillin. The encapsulation efficiency of vanillin was calculated using Equation (1):(1)$$wt\%=\frac{m}{M}\times 100\%$$
where *m* (mg) is the mass of vanillin encapsulated in the sample, and *M* (mg) is the mass of the carrier.

### 2.6. Release Performance Testing

The pH-responsive release behavior of vanillin in V@M, V@MM_2_, V@VM, and V@VMM was investigated by placing 40 mg of each sample in 40 mL water solutions with different pH values (7.0, 6.5, 5, and 3). After certain intervals, 1 mL of the supernatant was collected by centrifugation, and 1 mL of the corresponding pH water solution was used as a replacement. The amount of released vanillin over time was determined by measuring the absorbance of the collected supernatant.

The temperature-responsive release behavior of vanillin in V@M, V@MM_2_, V@VM, and V@VMM was investigated by placing 40 mg of each sample in 40 mL of deionized water at different temperatures (heating temperature (60 °C), room temperature, and refrigeration temperature (−4 °C)). At regular intervals, 1 mL of the supernatant was collected by centrifugation, and 1 mL of deionized water was used as a replacement. The released amount of vanillin was determined by measuring the absorbance of the collected supernatant. The cumulative release percentage (*CR%*) of vanillin was calculated using Equation (2):(2)$$CR\%=\frac{M_{r}}{M_{l}}\times 100\%$$
where $M_{r}$ is the cumulative release amount of vanillin, and $M_{l}$ is the loaded amount of vanillin.

## 3. Results and Discussion

ZIF-8 (M) and the secondary growth of ZIF-8-on-ZIF-8 (MM) were synthesized at room temperature following the methods reported previously in the literature. MM was synthesized using the as-synthesized ZIF-8 (M) as a template. To investigate the effect of different feed ratios on the growth of MM, M and different MM samples were prepared by controlling the feed ratios during synthesis. The feed ratios for the primary growth to secondary growth were set as 1:0.5, 1:1, 1:2, and 1:4, and they were named MM_0.5_, MM_1_, MM_2_, and MM_4_, respectively. The scanning electron microscopy (SEM) images showed that both the primary growth of ZIF-8 and the secondary growth of MM_0.5_ exhibited regular dodecahedral morphologies with smooth surfaces (Figure 1(a_1_,a_2_)). The particle size of M was approximately 100 nm, and MM_0.5_ had a slightly larger particle size than that of M. When the feed ratio was 1:2, MM_2_ also exhibited a dodecahedral structure. However, compared to M and MM_0.5_, MM_2_ had a less regular particle shape, with uneven sizes and slightly rougher surfaces (Figure 1(a_3_)). The particle size and morphology can affect the loading capacity of fragrance in the encapsulation system, system stability, and release performance of the fragrance. Figure 1b shows the particle size distribution of M and all MM samples. The primary growth of M had a narrow and uniform particle size distribution, with an average size of approximately 100 ± 50 nm. Interestingly, after the secondary growth of ZIF-8 using the MOF-on-MOF strategy, the particle size distribution of MM became wider. Moreover, as the core–shell feed ratio increased, the average particle size of MM gradually increased, indicating the formation of an additional layer of ZIF-8 on the surface of the primary ZIF-8 (M).

Figure 2 illustrates the X-ray diffraction (XRD) patterns of M and all MM samples. Since both the primary and secondary growths in MM involve the formation of ZIF-8, they exhibit the same topology and crystal parameters. Therefore, the XRD patterns of M and all MM samples are consistent and match the simulated XRD pattern of ZIF-8, indicating the successful synthesis of M and MM. To investigate the contribution of the specific surface area and pore structure of the carriers to the encapsulation of vanillin, nitrogen adsorption–desorption experiments were conducted on different carriers, and the isotherms are shown in Figure 3a. The results indicate that both M and MM samples with different feed ratios exhibit typical Type I isotherm characteristics, indicating the presence of predominantly microporous structures in the synthesized carriers. This is further confirmed by the pore size distribution plot (Figure 3b). As the feed ratio increased, the specific surface area and pore volume of the materials gradually decreased, which may be attributed to the increase in the particle sizes of the different carriers with the increasing feed ratio. The specific surface areas, micropore areas, and pore volumes of the different carriers are shown in Table 1. The specific surface area of M was 1473.9 m^2^ g^−1^, and the micropore volume was 0.710 m^2^ g^−1^. However, the specific surface area of MM_2_ decreased to 1237.9 m^2^ g^−1^, and the micropore volume decreased to 0.596 m^2^ g^−1^. The pore size distribution plot shows that the pore size distribution trend of the different MM carriers was similar to that of M, with two types of micropores distributed in the ranges of 4–6 nm and 8–9 nm. This confirms that the MM samples synthesized using this experimental method maintained the same pore structure as ZIF-8, suggesting the possibility of a double-layer core–shell structure in the prepared MM.

To verify the hypothesis that the secondary growth of the carrier MM forms another layer of ZIF-8 on the surface of ZIF-8, TEM characterization was performed on the different carriers, and the results are shown in Figure 4. The TEM images of M and MM_0.5_ show a regular hexagonal shape, which is a typical morphology of ZIF-8. Compared to the single-layer, regular dodecahedral structure of M (Figure 4a), the TEM image of MM_0.5_ clearly shows a double-layer structure. Combined with the SEM image of MM_0.5_, it can be concluded that MM_0.5_ has a regular double-layer core–shell structure (Figure 4b). Further observation of the morphology of MM_2_ using high-resolution transmission electron microscopy (HRTEM) (Figure 4c) reveals a decrease in regularity, consistent with the SEM results. In the high-resolution TEM image of MM_2_, a clear contour line can be observed, confirming the double-layer core–shell structure of MM_2_. To observe the structure of MM_2_ more intuitively, Fe was doped into the primary ZIF-8 at a ratio of 10:0.5 to synthesize the Fe-ZIF-8-on-ZIF-8 structure, and the distribution of elements in MM_2_ was observed. As shown in Figure 4d, Zn elements are uniformly dispersed throughout the entire MOF structure, while Fe elements, because of the low doping ratio, do not appear as distinct shapes. However, it can be seen that the density of Fe in the middle part of MM_2_ is higher than the edge part, which is sufficient to prove the double-layer core–shell structure of MM_2_.

The dynamic adsorption method was used to encapsulate vanillin in the pores of the single-layer M and double-layer MM, with MM_2_ as the representative of the double-layer ZIF-8-on-ZIF-8 structure, hereinafter referred to as MM. Vanillin was soaked into the pores of M and MM using ethanol as the solvent. Figure 5a shows the XRD spectra of M and MM after adsorption of vanillin. The results show that the XRD patterns of M and MM remained unchanged after the encapsulation of vanillin, indicating that the structures of M and MM did not change during the adsorption process. M and MM both exhibited stable rigid structures with strong structural stability. The effective encapsulation of vanillin was confirmed by infrared analysis. The infrared spectrum, shown in Figure 5b, reveals characteristic peaks of vanillin, carrier M, and the vanillin-adsorbed MM sample (V@MM). In the infrared spectrum of V@MM, peaks corresponding to the hydroxyl stretching vibration of vanillin (3180 cm^−1^), C=O stretching vibration (2840, 1670 cm^−1^), benzene-ring stretching vibration (1508 cm^−1^), and the 1,2,4-substituted peaks of the benzene ring (857, 815 cm^−1^) could be observed, confirming the successful encapsulation of vanillin in the carrier. Compared to the characteristic peaks of vanillin itself, such as the hydroxyl stretching vibration peak (3190 cm^−1^) and C=O stretching vibration peak (2860, 1660 cm^−1^), there is a shift in the peak position when vanillin was adsorbed in the carrier MM. This shift may be due to interactions between vanillin and the imidazole ring in MM, which effectively load vanillin into the pores of MM.

An electrostatic interaction is a common mechanism in solution adsorption. To observe the electrostatic interaction between vanillin and carriers M and MM, the zeta potentials of the carrier materials before and after the adsorption of vanillin were measured under neutral conditions, as shown in Figure 5c. The results show that the zeta potential of vanillin in ethanol solution was negative (−1.83 mV), while the zeta potentials of M (10.67 mV) and MM (13.43 mV) were positive, laying the foundation for the electrostatic interaction between vanillin and the carrier. Furthermore, vanillin contains electron-withdrawing groups (-OH), while ZIF-8 contains electron-donating groups (-CH_3_), further confirming the occurrence of electrostatic interaction between vanillin and ZIF-8. The absolute value of the zeta potential of MM was higher than that of M, indicating that compared to the M system, the MM system has greater stability, which is more favorable for the encapsulation and controlled release of vanillin. After the adsorption of vanillin, the zeta potential of V@M decreased to 8.57 mV, while the zeta potential of V@MM decreased to 8.08 mV. The higher potential difference indicates that the double-layer carrier MM has stronger electrostatic interaction with vanillin than the single-layer carrier M, which may be the main reason for the higher encapsulation capacity of double-layer MM compared to single-layer M.

The changes in thermal stability of the carrier and adsorbed fragrance before and after adsorption were studied by thermogravimetric analysis. Figure 5d shows that the V@MM sample had a greater weight loss at 1000 °C compared to pure ZIF-8. This behavior is due to the combination of the decomposition of vanillin and carrier MM during the analysis of the V@MM sample. Vanillin melts at 80 °C and decomposes at 160 °C, with almost complete evaporation at 240 °C. When heated to 1000 °C under nitrogen at the same rate, the carrier M started to decompose at approximately 375 °C, and the remaining mass at 1000 °C was approximately 51 wt% of the initial mass. On the other hand, the remaining mass of the V@MM sample was approximately 40 wt% of the initial mass. The difference in the remaining mass between the V@MM sample and pure M was 9 wt%, lower than the expected mass, which may be due to interactions between vanillin and the carrier. At the same time, the weight loss curve of V@MM was almost synchronized with M until about 550 °C, and then the weight loss of the carrier MM was significantly higher than that of the pure carrier M. It is speculated that the loss of vanillin in the V@MM sample may occur at around 550 °C, indicating that the presence of carrier MM greatly improves the thermal stability of vanillin. To observe the morphological changes in MM after adsorption of vanillin, SEM characterization was performed on the V@MM sample. The SEM image shown in Figure 5e reveals that the morphology of MM remained unchanged after the adsorption of vanillin, indicating that the encapsulation process did not cause significant morphological changes in the carrier.

Overall, the dynamic adsorption method successfully encapsulated vanillin in the pores of the single-layer M and double-layer MM. The encapsulation of vanillin was confirmed by XRD and infrared analysis. The electrostatic interaction between vanillin and the carrier was observed through zeta potential measurements. The thermal stability of vanillin was improved by the presence of the carrier MM. The morphology of MM remained unchanged after the adsorption of vanillin. These results demonstrate the effectiveness of the dynamic adsorption method for encapsulating fragrances in metal–organic frameworks.

The stability of the fragrance carrier is crucial for the encapsulation and protection of fragrances. The thermodynamic stability of the V@ZIF-8 system was studied using DFT calculations. The optimized model and computational results are shown in Figure 6a. The adsorption energy of vanillin in the ZIF-8 was −0.64 eV, indicating a strong interaction between vanillin and the ZIF-8 carrier. This interaction suggests that the adsorption of vanillin in ZIF-8 is thermodynamically stable. Additionally, the charge density analysis (Figure 6b) shows that there may be electron transfer between the O atom in vanillin and the H atom on the imidazole ring of ZIF-8. This further confirms the interaction between vanillin and ZIF-8.

To investigate the effect of the initial concentration of vanillin on its encapsulation capacity, a monolayer of ZIF-8 was used as the carrier. Vanillin solutions of different concentrations were adsorbed for the same duration, and the vanillin loading at different concentrations was compared. Considering that vanillin contains aromatic rings and double bonds, the concentration of vanillin in the carrier can be quickly and conveniently measured using UV absorption spectroscopy. Figure 7a shows the UV absorption spectra of vanillin, carrier M, and digested MM and V@MM samples. The results indicate that vanillin exhibits the strongest UV absorption peak at 280 nm. No peak was observed at this wavelength for the carrier MM before and after digestion with hydrochloric acid, while a UV absorption peak appeared at 280 nm for the digested V@MM sample, confirming the rationality of using UV absorption spectroscopy at 280 nm to determine the encapsulation of vanillin in the V@MM system. To accurately quantify the analysis, the absorbance at 280 nm was measured for vanillin solutions of different concentrations. Figure 7b shows a good linear relationship. The total adsorption and internal surface adsorption of the monolayer carrier M for vanillin at different initial concentrations were measured (Figure 7c). The adsorption results show that the total adsorption increased with the increase in the initial concentration of vanillin. However, the internal surface adsorption reached a maximum when the initial concentration of vanillin reached 1 mg L^−1^. When the initial concentration of vanillin continued to increase, the internal surface adsorption remained almost constant. This suggests that when the initial concentration of vanillin is 1 mg L^−1^, adsorption saturation is achieved. Higher concentrations are not conducive to the competitive adsorption of vanillin with the solvent ethanol. Most of the vanillin was more easily adsorbed on the external surface of carrier M, with only about 6 wt% of the vanillin loaded in the pores and internal surface of carrier M.

To investigate the encapsulation capacity of different carriers for vanillin, different carriers prepared in this experiment were added to a vanillin–ethanol solution with an initial concentration of 1 mg L^−1^ for adsorption and encapsulation. Figure 8 shows a comparison of the vanillin loading by the different carriers. The results indicate that under the same conditions, MM_2_ achieved the highest vanillin loading (both total adsorption and internal surface adsorption). As the feed ratio of the core–shell increased, the vanillin loading showed an increasing trend. The internal surface adsorption of MM_2_ for vanillin was 10.89 wt%, nearly twice that of the monolayer M (6.17 wt%). This is mainly attributed to the strong electrostatic interaction between the carrier MM_2_ and the guest molecule vanillin. However, the encapsulation capacity of MM_4_ for vanillin decreased. This may be because the particle size is larger, resulting in a decrease in the surface free energy, which reduces the adsorption affinity and makes it less favorable for adsorption.

To further increase the loading of vanillin, the in situ encapsulation + adsorption (secondary encapsulation) method was used with the monolayer M and double-layer MM for the encapsulation of vanillin. In situ encapsulation refers to the addition of a certain concentration of vanillin during the synthesis of the carrier, allowing it to be encapsulated in the pores and closed pores of the carrier along with the formation of the coordination structure of the carrier. The total adsorption and internal surface adsorption of vanillin obtained in this way are denoted as TA-1 and ISA-1, respectively. The samples obtained after in situ encapsulation were further subjected to dynamic adsorption for secondary encapsulation, and the final total adsorption and internal surface adsorption of vanillin are denoted as TA-2 and ISA-2, respectively.

The results of the encapsulation of vanillin using different methods are shown in Figure 9. The vanillin loading obtained through in situ encapsulation was relatively low for all three different carriers, VM, VMM, and VMVM. This may be because encapsulation occurs before the carrier has fully formed a specific coordination structure, so a large amount of vanillin cannot be securely trapped in the pores. Among them, VMM achieved the highest TA-1 and ISA-1, which were 6.35 wt% and 2.07 wt%, respectively, almost twice that of VM (TA-1: 3.40 wt%; ISA-1: 1.46 wt%). The reason for this result may be that in the VMM, vanillin was not only encapsulated in the pores of the ZIF-8 but also trapped between the core and shell of the double-layer ZIF-8. The vanillin loading in VMM was also higher than that in the VMVM, with TA-1 and ISA-1 of the VMVM being 3.64 wt% and 1.80 wt%, respectively. A possible reason is that for VMM, in situ encapsulation mainly occurs during the formation of the shell, and at this time, a complete monolayer ZIF-8 has been formed. This means that before in situ encapsulation occurs, the core structure can adsorb a portion of vanillin inside the pores and on the surface of ZIF-8, which increases the loading of vanillin in VMM (TA-1 and ISA-1). However, for VMVM, the encapsulation of vanillin occurs during the formation of both the core and shell. At this time, the carrier has not fully formed a stable coordination structure, resulting in a significant decrease in the encapsulation capacity. When the carriers obtained after in situ encapsulation were subjected to secondary adsorption, they had already formed stable porous materials, resulting in a significant increase in the loading of vanillin. The TA-2 and ISA-2 of VMM reached 30.09 wt% and 11.26 wt%, respectively, which were higher than the loading achieved through adsorption only (Appendix A). The VM and VMVM also showed a significant increase in loading after secondary adsorption. This is because in situ encapsulation can trap vanillin in regions that are not easily accessible by adsorption methods, such as between the core and shell, as well as in the closed pores of the carrier. It is worth noting that the vanillin loading achieved through adsorption methods reached nearly 80% in different encapsulation methods.

ZIF-8 has been proven to be an ideal pH-responsive drug delivery carrier. Under neutral or alkaline conditions, the N atoms in the ligand 2-methylimidazole become negatively charged due to deprotonation, and they coordinate with positively charged zinc ions to form the ZIF-8 structure. However, under acidic conditions, 2-methylimidazole is protonated and forms weak coordination with zinc ions, accelerating the collapse of the ZIF-8 structure and leading to the release of guest molecules [22]. The pH-responsive release performance of the vanillin encapsulation systems prepared using different carriers and encapsulation methods was tested, and the results are shown in Figure 10. Figure 10a,b show the release behavior of vanillin encapsulated by the carrier M and MM through adsorption under different pH conditions. A secondary kinetic model was used to fit the raw data. From Appendix A, it can be seen that the secondary kinetic model can better describe the release behavior of vanillin, indicating that vanillin mainly chemically interacted with ZIF-8, which is consistent with the DFT calculations. The results show that M exhibited s burst release behavior within the first 4 h, and the release rate of vanillin increased with decreasing pH. This is because the structure of the ZIF-8 carrier collapses to varying degrees under different pH conditions, and the stronger the acidity, the higher the degree of collapse, leading to faster release of vanillin from the encapsulation system. Under pH 3 conditions, the released amount of vanillin reaches 95 wt% after 4 h, while under neutral conditions, the released amount is only 43 wt% after 4 h and reaches only 60 wt% after 48 h. For the carrier MM, the presence of the second layer of ZIF-8 as a controlled entity may extend the kinetic equilibrium’s duration for the release of vanillin from 4 h (M) to 24 h (MM), greatly improving the burst release behavior of vanillin. Under neutral conditions, MM released 48 wt% of the vanillin after 48 h, but the absolute amount of vanillin released by MM under the same conditions was significantly higher than that of M, corresponding to the relatively higher vanillin loading of MM. Figure 10c,d show the kinetic release behavior of the encapsulation systems prepared using the in situ encapsulation + adsorption method for VM and VMM, respectively. VM exhibits the same burst release behavior as M, while VMM shows a similar sustained release behavior to MM. The double-layer ZIF-8-on-ZIF-8 had an excellent encapsulation amount, as well as delayed-release, compared to other systems (Appendix A). This further demonstrates that the second layer of MOF as a controlled entity in the double-layer carrier greatly improves the loading and kinetic equilibrium time of vanillin, achieving controlled release of guest molecules.
(3)$$q_{t}=\frac{{kq}_{e}^{2}t}{1+kq_{e}t}\times 100\%$$
where $q_{t}$ and $q_{e}$ are the vanillin cumulative release amount and equilibrium release amount at time t, respectively, and k is the rate constant.

Thermal-responsive release is a common mechanism for the release of guest molecules, where the interaction between the main and guest molecules weakens at high temperatures, leading to an accelerated release rate of the guest molecules. Figure 9 shows the release behavior of vanillin encapsulated by different carriers using different methods at various temperatures. Considering the temperature of commonly used fragrance products, refrigerated temperature (−4 °C), room temperature (RT), and heating temperature (60 °C) were chosen as the release temperatures for vanillin in this experiment. The results show that in the system in which the fragrance is encapsulated using the dynamic adsorption method (Figure 11a,b), the release of vanillin from M and MM followed a similar pattern (Appendix A), indicating that the change in temperature did not affect the carrier structure, and the carrier had good thermal stability, which is consistent with the thermogravimetric results. At 60 °C, the release rate of vanillin was the fastest, mainly because at higher temperatures, molecular motion becomes more active, weakening the interaction between vanillin molecules and the carrier, allowing vanillin molecules to escape from the carrier and enter the solution. However, compared to the single-layer M, MM still exhibited lower release rates. After 48 h of continuous release, the released amounts of MM at −4 °C, room temperature, and 60 °C were 43 wt%, 48 wt%, and 69 wt%, respectively, while for M, the released amounts were 54 wt%, 60 wt%, and 90 wt% at the same temperatures. Nevertheless, the absolute released amount of MM was still higher than that of M. These results demonstrate that in a temperature-responsive release system, the outer layer of MM improves the release rate of vanillin. Figure 11c,d show the release behavior of vanillin encapsulated by the VM and VMM carriers using the in situ encapsulation + dynamic adsorption method at different temperatures. The results show that the release pattern of VM was similar to that of M at different temperatures, while the release pattern of VMM was almost the same as that of MM. Interestingly, the released amount of VM was slightly lower than that of M after 48 h at each temperature, especially at low temperatures. The released amounts at −4 °C, room temperature, and 60 °C were 33 wt%, 49 wt%, and 88 wt% for VM, and it was 36 wt% for VMM at −4 °C, which is lower than the released amount of MM under the same conditions. This may be related to a small amount of vanillin being encapsulated in closed pores of the carrier material or in narrow areas that are not easily diffused during the in situ encapsulation process.

## 4. Conclusions

In this work, we synthesized single-layer ZIF-8 and double-layer ZIF-8-on-ZIF-8 structures through a simple room-temperature solvent method. The core–shell structure of ZIF-8-on-ZIF-8 was confirmed by characterization techniques such as SEM and TEM. Vanillin encapsulation was performed in both carriers using dynamic adsorption and in situ encapsulation + dynamic adsorption methods. The double-layer ZIF-8-on-ZIF-8 carrier exhibited a higher vanillin loading (25.68 wt%) compared to the single-layer carrier (10.89 wt%), which was nearly twice as high. This was mainly due to the strong electrostatic interaction between the double-layer carrier and vanillin. The pH-responsive and thermal-responsive release performances of vanillin in different encapsulation systems was tested. In the pH-responsive release experiments, at pH 3, 95 wt% of the vanillin was released in M after 4 h, while only 29.34 wt% was released in MM. In the thermally triggered experiments, at 60 °C for 48 h, the amount of vanillin released was 90.5 wt% in M, while 69.5 wt% was released in MM. In the double-layer carrier, the second layer of the MOF, as a controlled entity, greatly improved the vanillin loading and kinetic equilibrium time, reducing the release rate of vanillin and achieving the controlled release of guest molecules. In future work, the controlled release of various fragrances can be achieved by designing the MOF structure with a specific morphology and pore structures in different parts, encapsulating different fragrances accordingly. This can be achieved by regulating the interactions between different parts of the MOF and the fragrance molecules.

## Figures and Tables

**Figure 1 materials-17-01310-f001:**
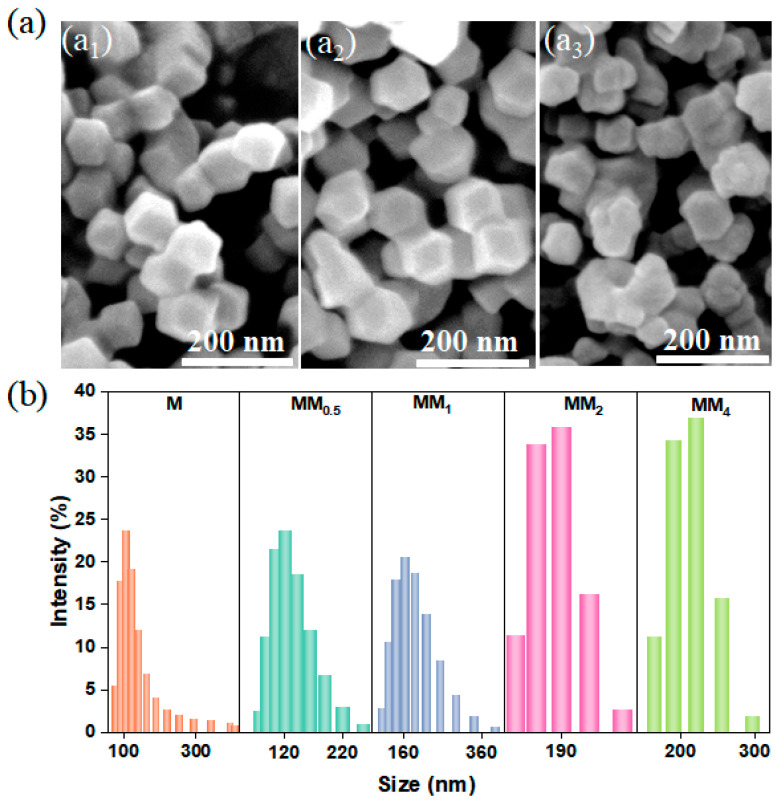
(**a**) SEM images of different MMs ((**a_1_**, **a_2_**, and **a_3_**) represent M, MM_0.5_, and MM_2_, respectively); (**b**) particle size distribution map (different samples were indicated by different colors in the figure).

**Figure 2 materials-17-01310-f002:**
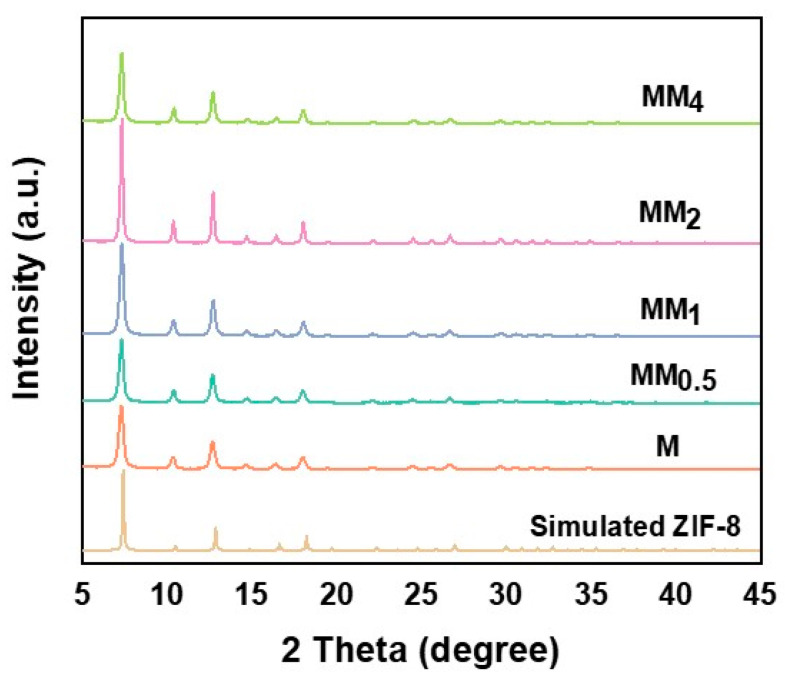
XRD patterns of the different MMs and Ms.

**Figure 3 materials-17-01310-f003:**
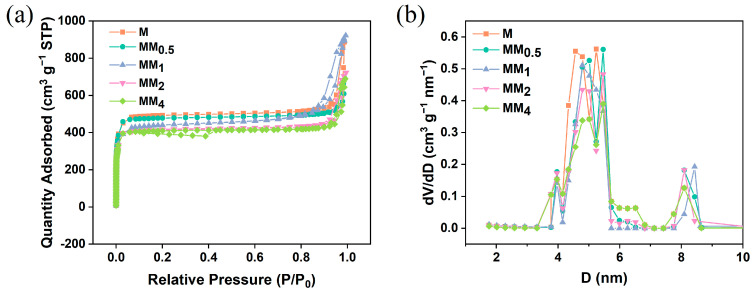
M and different MMs: (**a**) N_2_ adsorption and desorption isotherms; (**b**) aperture distribution map.

**Figure 4 materials-17-01310-f004:**
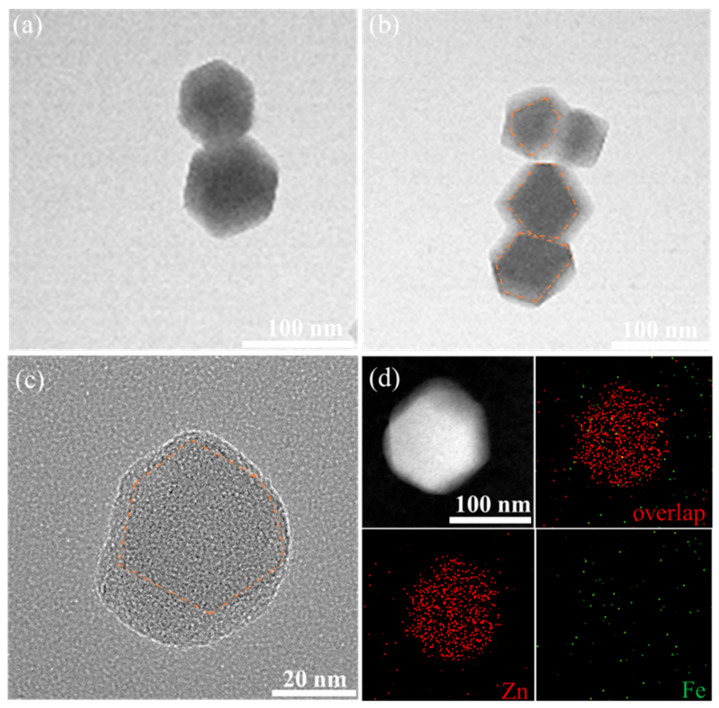
(**a**,**b**) TEM images of M and MM_0.5_; (**c**) TEM images of MM_2_ (the core is indicated by the orange outline); (**d**) HAADF-STEM and elemental analysis diagram of MM_2_.

**Figure 5 materials-17-01310-f005:**
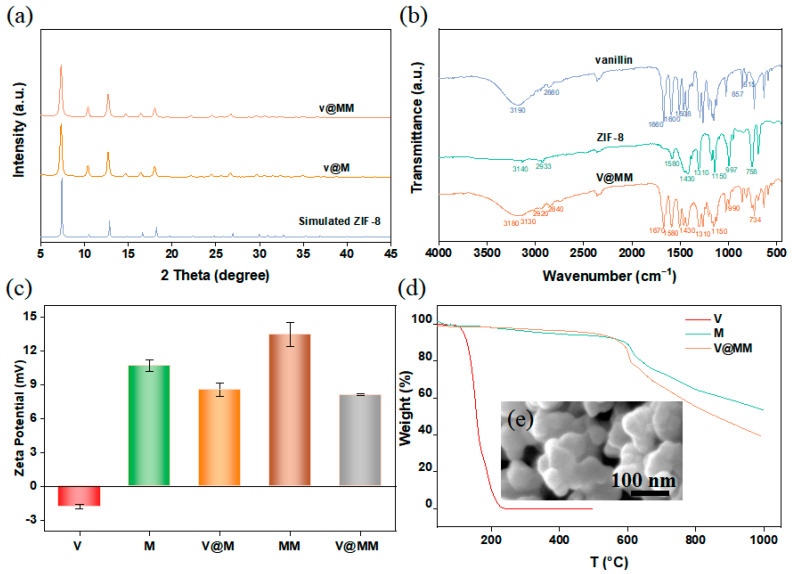
(**a**) XRD pattern of vanillin adsorbed on monolayer M and double-layer MM carrier; (**b**) infrared spectra; (**c**) zeta potentiogram; (**d**) thermogravigram; (**e**) SEM images.

**Figure 6 materials-17-01310-f006:**
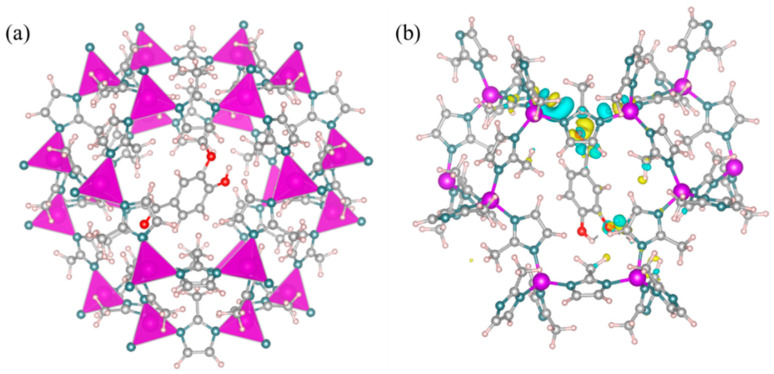
(**a**) Optimized V@ZIF-8 molecular configuration (gray for carbon, red for oxygen, blue for nitrogen, purple for zinc, and light pink for hydrogen); (**b**) differential charge density distribution for V@ZIF-8 (blue for decreasing charge density and yellow for increasing charge density).

**Figure 7 materials-17-01310-f007:**
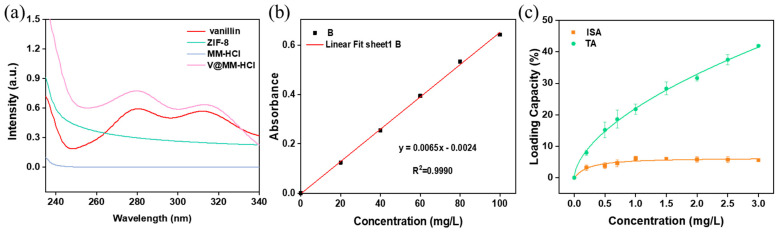
(**a**) Ultraviolet absorption spectra of vanillin and hydrochloric acid before and after digestion; (**b**) UV absorption linear curve of vanillin; (**c**) vanillin adsorption curve of carrier M at different initial concentrations.

**Figure 8 materials-17-01310-f008:**
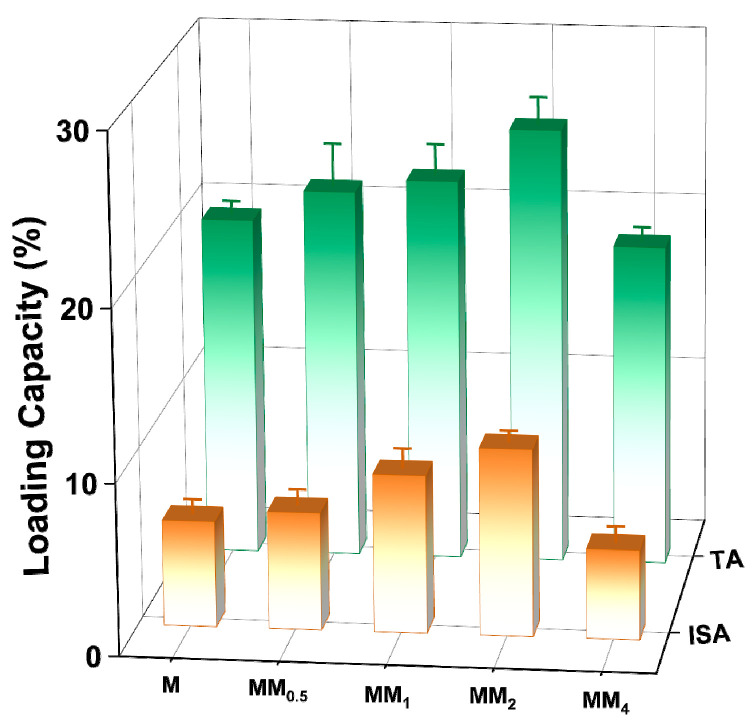
Loading capacity of vanillin adsorbed by different carriers (the total adsorption amount and surface adsorption amount was differentiated by different colors in the figure).

**Figure 9 materials-17-01310-f009:**
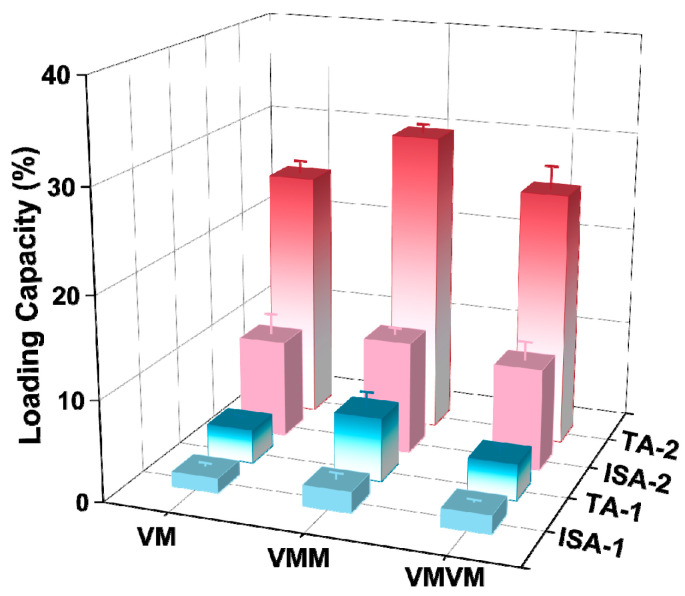
Comparison of vanillin loads in different encapsulation modes (the total adsorption amount and surface adsorption amount was differentiated by different colors in the figure).

**Figure 10 materials-17-01310-f010:**
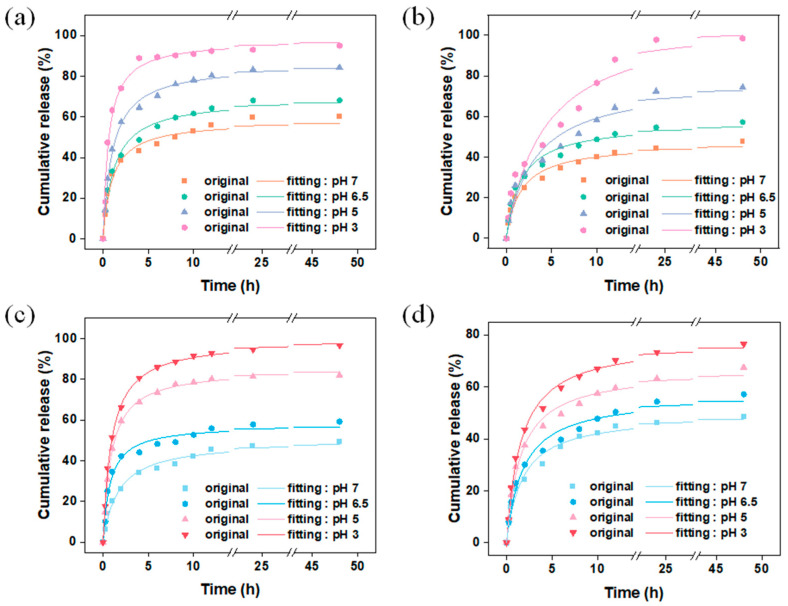
Carriers (**a**) M and (**b**) MM were used to encapsulate vanillin by dynamic adsorption; (**c**) release of vanillin in VM and (**d**) VMM under different pH conditions after in situ packaging and dynamic adsorption.

**Figure 11 materials-17-01310-f011:**
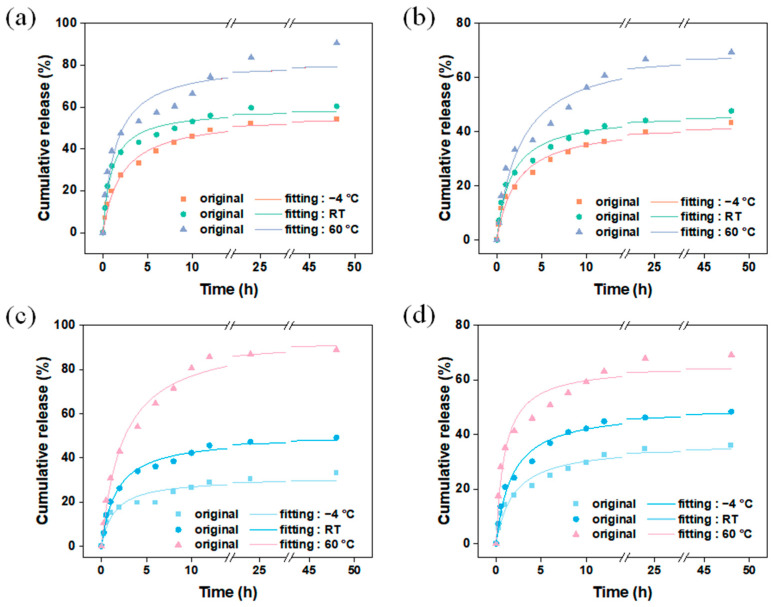
Carriers (**a**) M and (**b**) MM were used to encapsulate vanillin by dynamic adsorption; (**c**) release of vanillin in VM; (**d**) VMM at different temperatures after in situ packaging and dynamic adsorption.

**Table 1 materials-17-01310-t001:** BET data analysis of different vanillin encapsulation carriers.

Sample	BET Surface Area (m^2^ g^−1^)	Micropore Area (m^2^ g^−1^)	Micropore Volume (cm^3^ g^−1^)
M	1473.9	1353.80	0.71
MM_0.5_	1430.5	1344.00	0.70
MM_1_	1327.9	1142.56	0.60
MM_2_	1237.9	1136.36	0.60
MM_4_	1227.1	1144.63	0.60

## Data Availability

Data are contained within the article.

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
