# Peer review of "The Application of Bilayer Heterogeneous MOFs in pH and Heat-Triggered Systems for Controllable Fragrance Release"

_materials, 2024, doi:10.3390/ma17061310_

Round 1

Reviewer 1 Report

Comments and Suggestions for Authors

In this study, the authors present a novel dual-responsive release system for encapsulating fragrances, offering control through both pH and thermal triggers. The system employs ZIF-8 (M) and bilayer ZIF-8-on-ZIF-8 (MM) materials synthesized via a solvent method at room temperature.

Briefly compare the performance of this system to existing methods.

Add a sentence or two about future directions. 

Can you compare the performance of the system with that of a reference system (without double reaction)?

Figure 3 (a): Y axis “quantity absorbrd” should be “quantity absorbed”.

Compare the results of this work with other reports. Some recent important references can be incorporated in the manuscript.

Author Response

General comments: In this study, the authors present a novel dual-responsive release system for encapsulating fragrances, offering control through both pH and thermal triggers. The system employs ZIF-8 (M) and bilayer ZIF-8-on-ZIF-8 (MM) materials synthesized via a solvent method at room temperature.

Response: We thank the reviewer very much for the positive comments on our manuscript. We have carefully considered all the suggestions and made a point-to point response below.

Q1: Briefly compare the performance of this system to existing methods.

A1: Thanks for the kind suggestion. We have added relevant descriptions in the revised version.

Table R1 Comparison of encapsulation methods for different functional materials.

Methods

Materials

Performances

Characteristics

Ref.

Spray drying

Chitosan Encapsulated Orange Oil

90% encapsulation efficiency

Wide range of applications and easy to scale up, but limited by high temperatures

J. Agric. Food Chem. 2013, 61, 3311

Chemical polymerization

Epoxy resin encapsulated capsaicin

93% release in ethanol

Fast reaction rate and gentle reaction process, but lower mechanical integrity of the capsule

Polymer 2023, 285, 126369

Mesoporous adsorption

Mesoporous silica encapsulated curcumin

6.6% loading efficiency

Non-specific, but with low loading efficiency

Pharmaceutics 2019, 11, 430

Multiple adsorption

Bilayer ZIF-8-on-ZIF-8 encapsulated vanillin

25.68% loading efficiency

Dual-responsive release system with pH and thermal trigger control and high load efficiency

This work

Q2: Add a sentence or two about future directions.

A2: Thanks very much for the suggestion. We have added the relevant description in the conclusion section of the revised manuscript.

“In future work, the controlled release of various fragrances can be achieved by designing the MOF structure with specific morphology and pore structures in different parts and encapsulating different fragrances accordingly. This can be achieved by regulating the interactions between different parts of the MOF and the fragrance molecules.” (lines 516-520, page 15).

Q3: Can you compare the performance of the system with that of a reference system (without double reaction)?

A3: Thanks very much for this important question. In this work, we compared the encapsulation capacity as well as pH-responsive release and heat release properties of single-layer ZIF-8 and double-layer ZIF-8-on-ZIF-8 for vanillin. Meanwhile, to further improve the vanillin loading, the encapsulation and release performance of vanillin were tested by in situ encapsulation + adsorption (secondary encapsulation) with single-layer ZIF-8 and double-layer ZIF-8-on-ZIF-8, respectively. From Table R2, it can be seen that bilayer ZIF-8-on-ZIF-8 has higher vanillin loading capacity as well as superior slow-release ability compared to monolayer ZIF-8. To make the above results clearer, we have added the relevant performance comparison tables in the supporting information.

Table R2 Comparison of single-layer ZIF-8 and double-layer ZIF-8-on-ZIF-8 performance.

Samples

ISAC

TAC

CR (pH=3, 10 h)

CR (60 ℃, 48 h)

ZIF-8 (M)

6.17%

19.97%

91 wt%

90 wt%

ZIF-8-on-ZIF-8 (MM)

10.89%

25.68%

76 wt%

69 wt%

vanillin@ZIF-8 (VM)

9.80%

24.87%

91 wt%

88 wt%

vanillin@ZIF-8-on-ZIF-8 (VMM)

11.26%

30.09%

67 wt%

69 wt%

Q4: Figure 3 (a): Y axis “quantity absorbrd” should be “quantity absorbed”.

A4: Thanks very much for the correction. We have changed "absorbed" to "absorbed" in Figure 3(a) (line 244, page 16).

Figure R1. M and different MM (a) N2 adsorption and desorption isotherms; (b) Aperture distribution map.

Q5: Compare the results of this work with other reports. Some recent important references can be incorporated in the manuscript.

A5: Thanks very much for the important suggestion. We have added a table comparing the performance of this work with other reports in the supporting information.

Table R3 Comparison of encapsulation properties of functional materials with different matrixes.

Matrixes

Functional materials

Methods

Loading capacity

Cumulative release

Ref.

ZIF-8

Physcion

Adsorption

11.49 wt%

88.72 wt% (pH=5, 72 h)

Colloid. Surface. B. 2019, 182, 110364

ZIF-8@gallic acid@Fe

5-fluorouracil

Adsorption

24.77 wt%

54 wt% (pH=5, 12 h)

J. Drug Deliv. Sci. Tec. 2023, 87, 104878

ZIF-8

Gentiopicroside

Adsorption

10.77 wt%

81.31 wt% (pH=5, 50 h)

J. Drug Deliv. Sci. Tec. 2023, 84, 104530

Whey protein isolate

vanillin

Non-thermal spray-freeze-drying

39.22 wt%

——

Food Chem. 2015, 174, 16

ZIF-8

vanillin

Adsorption

19.97 wt%

84.42 wt% (pH=5, 48 h)

This work

ZIF-8-on-ZIF-8

vanillin

Adsorption

25.68 wt%

74.31 wt% (pH=5, 48 h)

This work

Reviewer 2 Report

Comments and Suggestions for Authors

The manuscript entitled “The application of bilayer heterogeneous MOFs in pH and heat-triggered systems for controllable fragrance release”, reports research on the preparation of ZIF-8 and ZIF-8 on ZIF-8 respectively single and double layer materials for the 'encapsulation of fragrances. The preparation and characterization of the samples obtained are reported in the manuscript. The authors report that the double-layer samples show greater fragrance retention than the single-layer sample under the same conditions. I believe that in its current version the manuscript cannot be considered for publication. Here are some suggestions for authors:

- In the introduction, greater in-depth analysis should be addressed to imidazolate zeolitic structures (ZIF), being the materials considered in this study: what are their characteristics? Have they been used in other similar research? No reference is reported.

- I believe that reading is not easy due to the many acronyms that are used to name the samples. The manuscript should be revised especially with regards to the expository aspect. Can't follow well. I suggest revising it, perhaps also including tables that summarize the compositions of the prepared samples.

In light of the above, I believe that the manuscript needs more revision before being considered for publication.

Author Response

General comments: The manuscript entitled “The application of bilayer heterogeneous MOFs in pH and heat-triggered systems for controllable fragrance release”, reports research on the preparation of ZIF-8 and ZIF-8 on ZIF-8 respectively single and double layer materials for the 'encapsulation of fragrances. The preparation and characterization of the samples obtained are reported in the manuscript. The authors report that the double-layer samples show greater fragrance retention than the single-layer sample under the same conditions. I believe that in its current version the manuscript cannot be considered for publication. In light of the above, I believe that the manuscript needs more revision before being considered for publication. Here are some suggestions for authors:

Response: We thank the reviewer very much for the comments on our manuscript. We have carefully considered all the suggestions and made a point-to point response below.

Q1: In the introduction, greater in-depth analysis should be addressed to imidazolate zeolitic structures (ZIF), being the materials considered in this study: what are their characteristics? Have they been used in other similar research? No reference is reported.

A1: Thanks very much for the suggestion. We have added a description in the Introduction section of the revised manuscript.

“ZIF-8, as a zeolitic imidazole framework material, is self-assembled from physiological system components, Zn2+, and 2-methylimidazole, and has excellent biocompatibility (ACS Nano 2014, 8, 2812). Meanwhile, ZIF-8 exhibits good thermal stability due to the strong interaction between the core metal ions and the nitrogen atoms in the coordination ligand (J. Taiwan Inst. Chem. E. 2023, 149, 104993). ZIF-8 has a high specific surface area, high porosity, controllable size, and easy synthesis, making it highly capable of loading functional materials (Colloid. Surface. B. 2019, 182, 110364). In addition, ZIF-8 has excellent acid responsiveness and can be stabilized in physiological aqueous environments, while the zeolite imidazolium ester skeleton collapses when the pH is weakly acidic (pH 5-6), which facilitates the control of the release of the functional materials by adjusting the pH (Dalton Trans. 2012, 41, 6906). Based on the above characteristics ZIF-8 is widely used in the biomedical field to encapsulate various functional materials, such as 5-fluorouracil (J. Drug Deliv. Sci. Tec. 2023, 87, 104878), gentiopicroside (J. Drug Deliv. Sci. Tec. 2023, 84, 104530), riboflavin-5-phosphate (Adv. Mater. 2022, 34, 2109865), 6-mercaptopurine (J. Drug Deliv. Sci. Tec. 2017, 41, 106), and metformin (Mater. Technol. 2022, 37,926).” (lines 64-75, page 2).

Q2: I believe that reading is not easy due to the many acronyms that are used to name the samples. The manuscript should be revised especially with regards to the expository aspect. Can't follow well. I suggest revising it, perhaps also including tables that summarize the compositions of the prepared samples.

A2: Thanks very much for the suggestion. To make the manuscript easier to read, we added a list of abbreviations to the supporting information to clarify what each abbreviation stands for, as well as the composition of the different samples.

Table R4 Full name corresponding to the abbreviation.

Abbreviations

Full name

MOFs

Metal-organic frameworks

ZIFs

Zeolitic imidazolate frameworks

2-MIm

2-methylimidazole

M

ZIF-8

MM

ZIF-8-on-ZIF-8

VM

Vanillin@ZIF-8

VMM

Vanillin@ZIF-8-on-ZIF-8

VMVM

Vanillin@ZIF-8-on-vanillin@ZIF-8

GGA

Generalized gradient approximation

PBE

Perdew-Burke-Ernzerhof

PAW

Projector augmented-wave

TAC

Total adsorption capacity

ISAC

Inner surface adsorption capacity

V@M

ZIF-8 adsorbed vanillin

V@MM2

ZIF-8-on-ZIF-8 (1:2) adsorbed vanillin

V@VM

Vanillin@ZIF-8 adsorbed vanillin

V@VMM

Vanillin@ZIF-8 adsorbed vanillin

XRD

X-ray diffraction

SEM

Scanning electron microscope

TEM

Transmission electron microscopy

HRTEM

High-resolution transmission electron microscopy

RT

Room temperature

Reviewer 3 Report

Comments and Suggestions for Authors

In this manuscript, the authors designed a bi-responsive release system with pH and thermal response control to apply and release fragrance encapsulation systems into ZIF-8 and bilayer ZIF-8-on-ZIF-8 materials. It is an interesting study, and I would like to recommend it for publication after carefully implementing the following suggestions and comments:

1. The author has done extensive research in the biomedical field, and it would be nice to add more relevant literature (line 58 &59)

2. Did the authors provide the trend lines in Figure 10 and Figure 11 through simple numerical analysis of the data? Quantitative data should be provided that analyzes the parameters and the applied model by applying an appropriate model, such as a first-order kinetics model, related to the release of encapsulated vanillin.

Author Response

General comments: In this manuscript, the authors designed a bi-responsive release system with pH and thermal response control to apply and release fragrance encapsulation systems into ZIF-8 and bilayer ZIF-8-on-ZIF-8 materials. It is an interesting study, and I would like to recommend it for publication after carefully implementing the following suggestions and comments:

Response: We thank the reviewer very much for the constructive suggestions on our manuscript. We have carefully considered all the suggestions and made all necessary revisions to the manuscript.

Q1: The author has done extensive research in the biomedical field, and it would be nice to add more relevant literature (line 58 &59)

A1: Thanks for the kind suggestion. We have added relevant descriptions in the revised version.

“MOFs have been extensively applied in biomedical applications, such as Zr-MOF-loaded naproxen as a carrier for specific intestinal delivery (Chem. Pap. 2023, 77, 3461), Mg-MOF to facilitate rapid drug metabolism (ACS Appl. Bio Mater. 2023, 6, 2477), and Ca-Sr-MOF-loaded tetracycline for antimicrobial use (ACS Omega 2023, 8, 41909). The excellent encapsulation properties of MOF materials make them a unique encapsulation system.” (lines 59-63, page 2).

Q2: Did the authors provide the trend lines in Figure 10 and Figure 11 through simple numerical analysis of the data? Quantitative data should be provided that analyzes the parameters and the applied model by applying an appropriate model, such as a first-order kinetics model, related to the release of encapsulated vanillin.

A2: Thanks very much for this important question. In Figures 10 and 11 the scatter points are the original data and the curves are the fitted trend lines, to make this easier to distinguish we have updated the legends in Figures 10 and 11 in the revised manuscript. We used a secondary kinetic model (Eq.1) to fit the original data by performing a simple numerical analysis of the original data as well as considering the electron transfer between vanillin and ZIF-8. From Tables R5 and R6, we can see that the secondary kinetic model can better describe the release behavior of vanillin, indicating that vanillin is mainly chemically interacting with ZIF-8. To clarify the above results we describe them in the revised manuscript and provide quantitative data in the supporting information (lines 435-438, page 12).

                                                                                                                (1)

Figure R2. Carrier (a) M and (b) MM were used to encapsulate vanillin by dynamic adsorption. (c) Release of vanillin in VM and (d) VMM under different pH conditions after in situ packaging and dynamic adsorption.

Figure R3. Carrier (a) M and (b) MM were used to encapsulate vanillin by dynamic adsorption. (c) Release of vanillin in VM and (d) VMM at different temperatures after in situ packaging and dynamic adsorption.

Table R5 Secondary kinetic fitting parameters for different samples at different pHs.

Samples

pH

3

5

6.5

7

M

qe

97.685

85.751

68.800

58.000

k

0.016

0.011

0.012

0.017

R2

0.987

0.996

0.989

0.982

MM

qe

108.688

77.363

56.534

46.541

k

0.002

0.004

0.011

0.013

R2

0.960

0.968

0.982

0.982

VM

qe

99.222

85.068

57.575

49.575

k

0.011

0.013

0.022

0.012

R2

0.999

0.998

0.981

0.989

VMM

qe

77.574

66.354

56.526

49.332

k

0.008

0.010

0.010

0.011

R2

0.995

0.991

0.983

0.985

Table R6 Secondary kinetic fitting parameters for different samples at different temperatures.

Samples

Temperature

-4 °C

RT

60 °C

M

qe

55.524

58.862

81.420

k

0.009

0.017

0.009

R2

0.991

0.983

0.933

MM

qe

42.786

46.541

70.644

k

0.011

0.013

0.005

R2

0.980

0.982

0.967

VM

qe

30.685

49.579

95.295

k

0.023

0.012

0.004

R2

0.940

0.989

0.988

VMM

qe

35.994

49.330

65.376

k

0.014

0.012

0.016

R2

0.971

0.985

0.959

Round 2

Reviewer 2 Report

Comments and Suggestions for Authors

The authors have improved the manuscript by reviewing some critical aspects and therefore I believe that the manuscript can be considered for publication.

Reviewer 3 Report

Comments and Suggestions for Authors

The authors have made all the improvements I asked them to make, and I recommend the manuscript for publication in its current state.